# Large-scale photonic chip based pulse interleaver for low-noise microwave generation

Zheru Qiu [1,2,6], Neetesh Singh [3,6] ✉, Yang Liu [1,2], Xinru Ji[1,2], Rui Ning Wang[1,2,5], Franz X. Kärtner [3,4] ✉ & Tobias Kippenberg [1,2] ✉

Optically generated microwaves exhibit unprecedented low noise, benefiting applications such as communications, radar, instrumentation, and metrology. To date, the purest microwave signals are produced using optical frequency division with femtosecond mode-locked lasers. However, their typical repetition rates of hundreds of MHz require multiplication methods to reach the microwave domain. Here, we introduce a miniaturized photonic integrated circuit-based interleaver, achieving a 64-fold multiplication of the repetition rate from 216 MHz to 14 GHz in Ku-Band. With the interleaver, the generated microwave power was improved by 35 dB, with a phase noise floor reduced by more than 10 folds by alleviating photodetector saturation. Based on a low-loss and high-density $Si_3N_4$ waveguides, six cascaded stages of Mach-Zehnder interferometers with optical delay lines up to 33 centimeters long are fully integrated into a compact chip. Our result can significantly reduce the cost and footprint of mode-locked-laser-based microwave generation, enabling field deployment in aerospace and communication applications.

Low-phase-noise microwaves are crucial for a wide range of applications, including communications[1,2], radar[3,4], metrology[5], and instrumentation. In radar systems, elevated phase noise degrades the ability to distinguish between targets and clutter while also reducing ranging precision[6]. In communication systems, phase noise in the local oscillator can introduce random rotations in the received signal constellation[7] and cause interference in adjacent channels by spectrum regrowth[8]. Additionally, in test equipment that employs high-speed analog-to-digital converters, such as oscilloscopes and signal analyzers, a low-noise clock is essential for achieving a high dynamic range[9].

Currently, state-of-the-art low-noise microwave generation deployed in practical applications primarily relies on compact dielectric resonant oscillators (DROs), which have been engineered and refined extensively over the past decades. In the laboratory environment, exceptionally low-noise microwave signals can be generated by optical means[10,11]. In particular, low phase noise aspect of an ultra-stable optical reference can be transferred down to the microwave regime with a powerful technique known as optical frequency division (OFD), which has resulted in low-phase-noise microwaves reaching below -170 dBc/Hz at offsets >10 kHz[10,12]. In this technique, a solid-state or fiber-based mode-locked laser (MLL) is usually stabilized by locking a comb line to an ultra-low noise optical reference. The stabilized repetition rate is then converted to a high-purity electronic signal by photodetection.

A challenge in practical applications of OFD is the gap between the pulse repetition rate and the microwave frequencies desired for utilization. Most important applications such as communication systems, radar, and instrumentation require high-purity microwaves in the range of a few to tens of GHz. At the same time, the repetition rate of

[1]Swiss Federal Institute of Technology Lausanne (EPFL), Lausanne, Switzerland. [2]Center for Quantum Science and Engineering, EPFL, Lausanne, Switzerland. [3]Center for Free-Electron Laser Science, Deutsches Elektronen-Synchrotron, Hamburg, Germany. [4]Department of Physics, Universität Hamburg, Hamburg, Germany. [5]Present address: Luxtelligence SA, Lausanne, Switzerland. [6]These authors contributed equally: Zheru Qiu, Neetesh Singh. ✉e-mail: neetesh.singh@desy.de; franz.kaertner@desy.de; tobias.kippenberg@epfl.ch

the low-noise MLLs typically falls in the range of a few hundred MHz. The short photocurrent pulses naturally contain high-order harmonics of the repetition rate, which can be used to bridge the gap in frequency. However, the signal-to-noise ratio (SNR) of the generated microwave is limited by the photocurrent saturation in the photodiodes (PDs), even with state-of-the-art modified uni-traveling-carrier (MUTC) PDs[13]. The PDs saturate by the incident of the high-peak-power femtosecond (fs)-pulses from MLLs, slowing down the response and thus limiting the achievable power of high-order harmonics. While there is an elegant technique that bypasses this challenge by phase-locking to the rising edge of the photocurrent pulses[14], it necessitates careful control of laser intensity noises and high system complexity. Soliton microcombs, on the other hand, offer a high-repetition-rate pulse train[15], but the currently achievable noise performance even with delicate feedback stabilization[8] is not yet comparable with conventional MLLs.

A common strategy to alleviate this problem is multiplying the optical pulse repetition rate using mode-filtering cavities[16,17] or pulse interleavers[18]. The pulse train with multiplied repetition rates has a lower peak power at the PD, which increases the generated microwave power by reducing saturation and improving the signal-to-noise ratio limited phase noise floor. These pulse rate multipliers are used in the experiments demonstrating the generation of the purest microwave signal[12] and commercial optical microwave generation systems[19]. However, the mode filtering cavities will reduce the available optical power and require servo locking. The fiber-based and free-space interleavers have a large footprint (Fig. 1b), are prone to mechanical noise and temperature fluctuations, and necessitate careful adjustment for achieving desired delay lengths[20].

In this work, we demonstrate a miniaturized 6-stage pulse interleaver on a low-loss silicon nitride photonic integrated circuit (Fig. 1c) fabricated at wafer scale. By incorporating up to 33 cm long low-loss delay lines, this chip can multiply the repetition rate $f_{rep} = 216.7$ MHz of a table-top 1550 nm MLL by $2^6$ times. The device footprint of 8.5 mm × 1.7 mm device (Fig. 1d) is significantly smaller than fiber-based or free-space counterparts and is more than one order of magnitude smaller than the reported 4-stage low-index silica waveguide-based mode-filter[21]. This device allows for low-cost mass production of compact ultra-low-phase-noise photonic microwave sources well suited for miniaturized solid-state MLLs or the future integration with chip-based MLLs[22,23].

## Results

We adopt a cascaded unbalanced Mach-Zehnder interferometer (MZI) design (Fig. 1a) for the interleaver. In the $n$th stage of an ideal interleaver, each input pulse is equally divided into two halves by a coupler. The pulse train in the longer MZI arm is delayed by $1/(2^n f_{rep})$ and is subsequently combined with the pulse in the other half at the output coupler of each stage. In the frequency domain, this pulse interleaving process modifies the relative phase of the comb lines of the MLL pulse

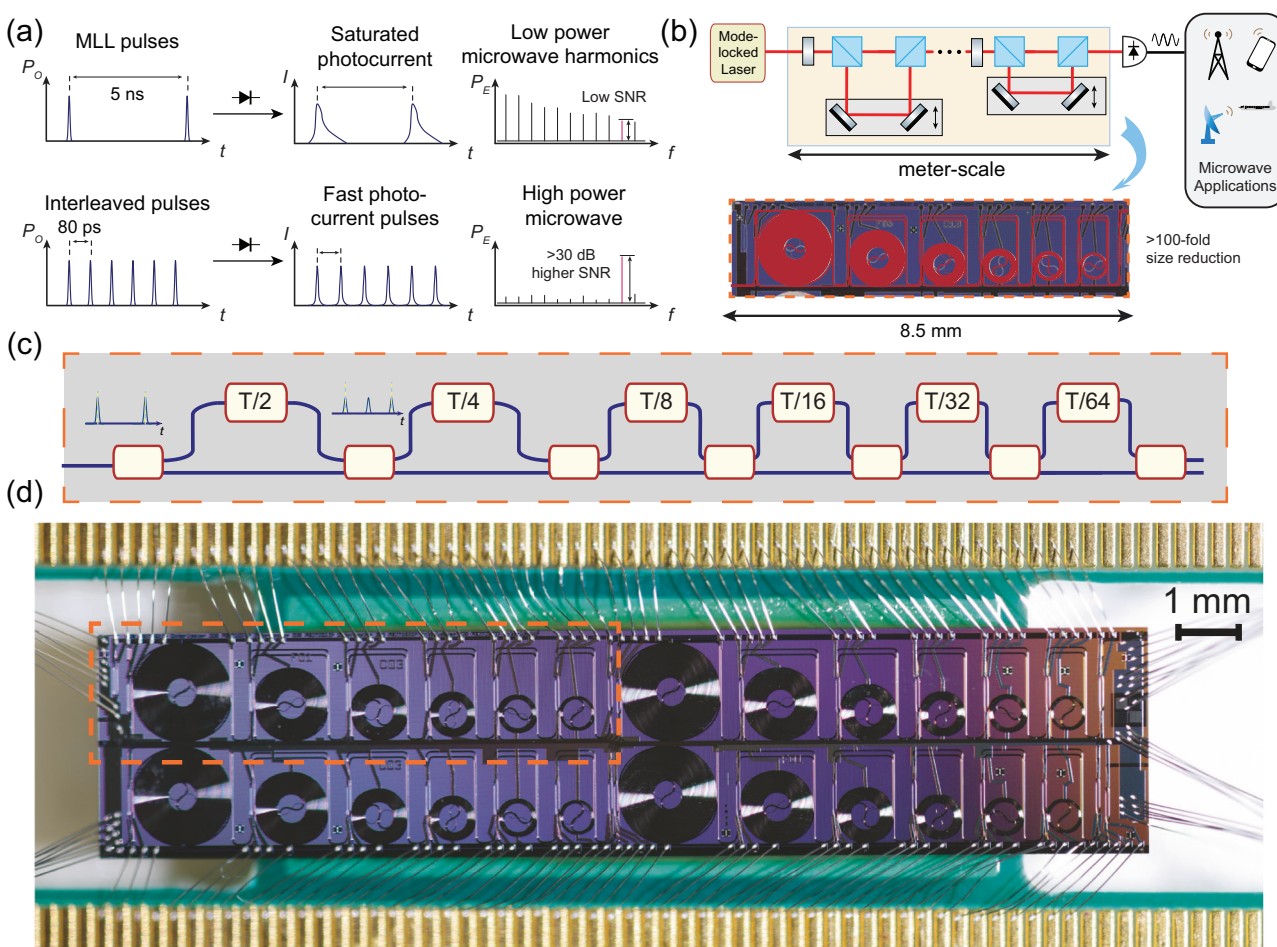

**Fig. 1 | Large-scale photonic integrated circuit-based interleaver for low-noise microwave generation. a** Illustration of the photonic microwave generation process with a mode-locked laser and the improvement of microwave power and SNR through pulse interleaving. **b** Size reduction of the pulse interleaver by replacing the free space optics with the on-chip components in a photonic microwave generation system. Inset is the optical micrograph of the fabricated Si$_3$N$_4$ interleaver chip, overlaid with the schematic of the Si$_3$N$_4$ waveguide. **c** Illustration of the working principle of the cascaded unbalanced Mach-Zehnder interferometer pulse interleaver. **d** Top view image of the 6-stage interleaver chip mounted on a printed circuit board for breaking out the heater connections.

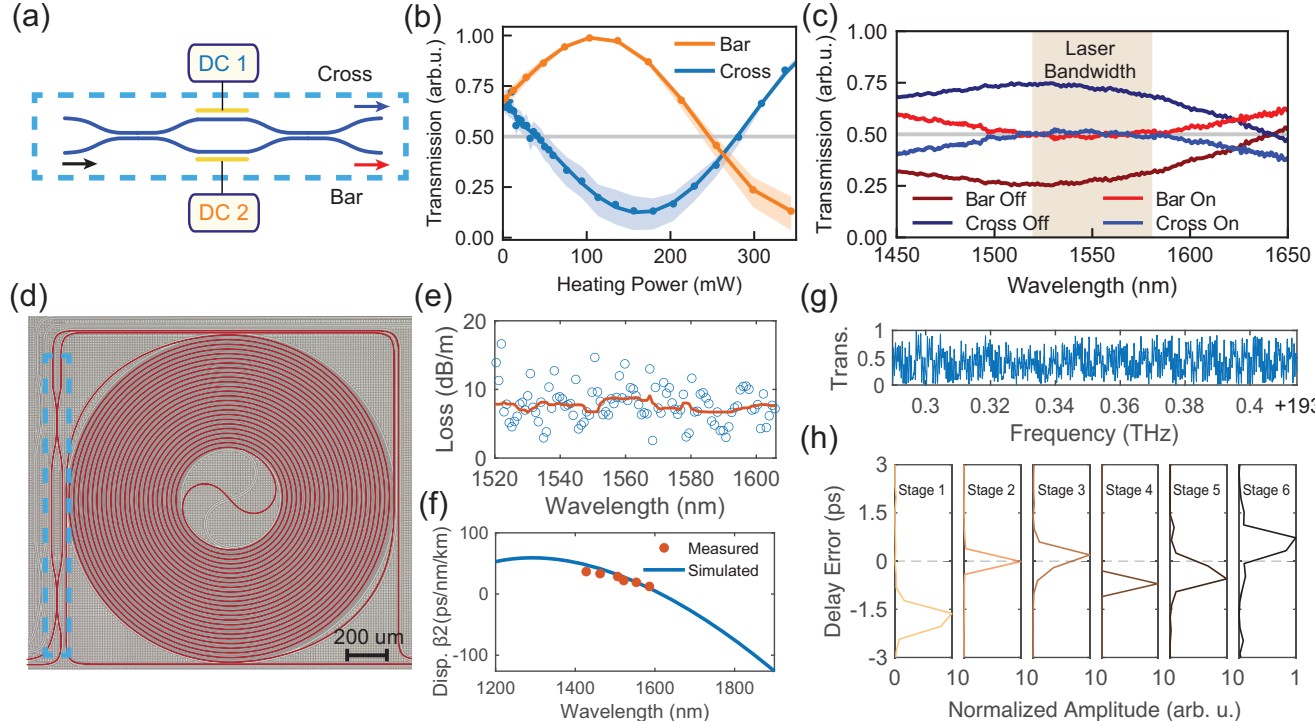

**Fig. 2 | Characterization of the Si₃N₄ photonic chip based pulse interleaver.**
**a** Schematic of the balanced MZI tunable coupler implemented with two 50:50 waveguide directional couplers. DC: direct current. **b** Optical transmission through the bar port as a function of the heating power applied to either arm at 1550 nm. The shaded area shows the minimum to maximum range of the transmission within 1550 ± 10 nm. **c** Normalized transmission spectrum of the tunable coupler when cold and having 187 mW heating applied to one arm. **d** Optical micrograph of a single interleaver stage, highlighting the tunable coupler in (**a**). **e** Waveguide propagation loss of the delay line section, characterized by the Lorentzian fit of ring resonator resonances, which gives the intrinsic loss rate. Circles represent individual resonances, and the red line is the moving median. **f** Simulated and characterized waveguide dispersion of the long delay line, described as $D = -2\pi c \beta_2 / \lambda^2$, where $c$ is the speed of light and $\beta_2$ is the group velocity dispersion parameter. **g** Normalized transmission spectrum through the 6-stage interleaver near 193 THz, showing a part of the interference fringes used in the delay length characterization. **h** Time delay error characterized by the Fourier transformation of the optical transmission spectrum, zoomed in to the relevant frequency component indicating the time delay errors of each delay line.

train, while the amplitudes remain unchanged. Therefore, for an ideal interleaver, there is no energy loss when summing up the pulse energies at both outputs. The two outputs can be simultaneously utilized by separate photodetection and coherently combining the generated microwave. This is in stark difference from the approaches that filter out modes[17,21] to increase the frequency interval, where the total energy of the pulse train is reduced to $1/m$ for a $m$-times repetition rate multiplication.

To compensate for the fabrication uncertainty in the coupler splitting ratio and unbalanced delay line losses for each interleaver stage, we employ balanced MZI tunable couplers composed of two 50:50 directional couplers and two arms integrated with Ti/Pt microheaters (Fig. 2a). The adjustment of the splitting ratio is achieved by applying different phase shifts to the two arms (Fig. 2b). Figure 2c shows the normalized transmission spectrum through the fabricated MZI directional couplers, tested with an in-house comb-assisted laser-sweeping-based optical network analyzer[24]. The transmission can be tuned to 50 ± 4% within the 1550 ± 30 nm range, ensuring equal splitting of all the spectral components of the MLL comb. We anticipate that the use of thermal isolation trenches for future development can enhance tuning efficiency[25]. The Si₃N₄ photonic integrated circuits are fabricated with the photonic damascene process[26]. The delay line waveguides are designed to have a 1.5 μm × 700 nm cross-section which is narrower than previous works to reduce the coupling to high-order modes; as a trade-off, the waveguide scattering loss is higher than wider waveguides. The total waveguide propagation loss is in the range of 8 to 10 dB/m (Fig. 2). This low waveguide loss only contributes to a total on-chip optical loss of 2.2 dB,

allowing for minimal degradation in SNR. Figure 2f shows the simulated and measured (Supplementary Note 1) waveguide dispersion, respectively. The signal wavelength range lies in the weak anomalous dispersion regime.

The lithography can define precise optical delay length for the interleaver, which is critical to ensuring low errors of time delay that can otherwise corrupt the regular timing between the pulses, decrease the generated microwave power and negate the phase noise reduction by shot noise correlations of the short pulses[27,28]. On the interleaver chip, the time delay is determined by the waveguide physical length and group index, which are both precisely defined by deep ultraviolet lithography. We characterized the time delay of each interleaver stage on fabricated chips by performing a Fourier transform of the transmission spectrum of the interleaver (Supplementary Note 2). Figure 2(h) shows the longest delay time of 2308 ps (corresponding to 33 cm) with a maximum timing error of <2 ps (0.09%). For shorter delay lengths, the delay time errors are smaller and close to the resolution limit of this characterization technique (0.4 ps, limited by the analysis wavelength range of 1540–1560 nm). In addition, we characterized the variation of the delay length across the 4-inch wafer, where we find that the variance between the 9 stepping fields is around 0.57 ps (0.025%). This small delay error is comparable to or even better than the hand-polished fiber delay lines, and the high consistency across the wafer allows for volume fabrication with high yield. We note that inserting more stages with shorter delay lengths of a few mm or hundreds of μm is difficult with fiber delay lines but can be simply implemented on a chip. Using photonic integrated circuits, the repetition rate of the generated pulse train can be easily extended to >100 GHz[29], enabling

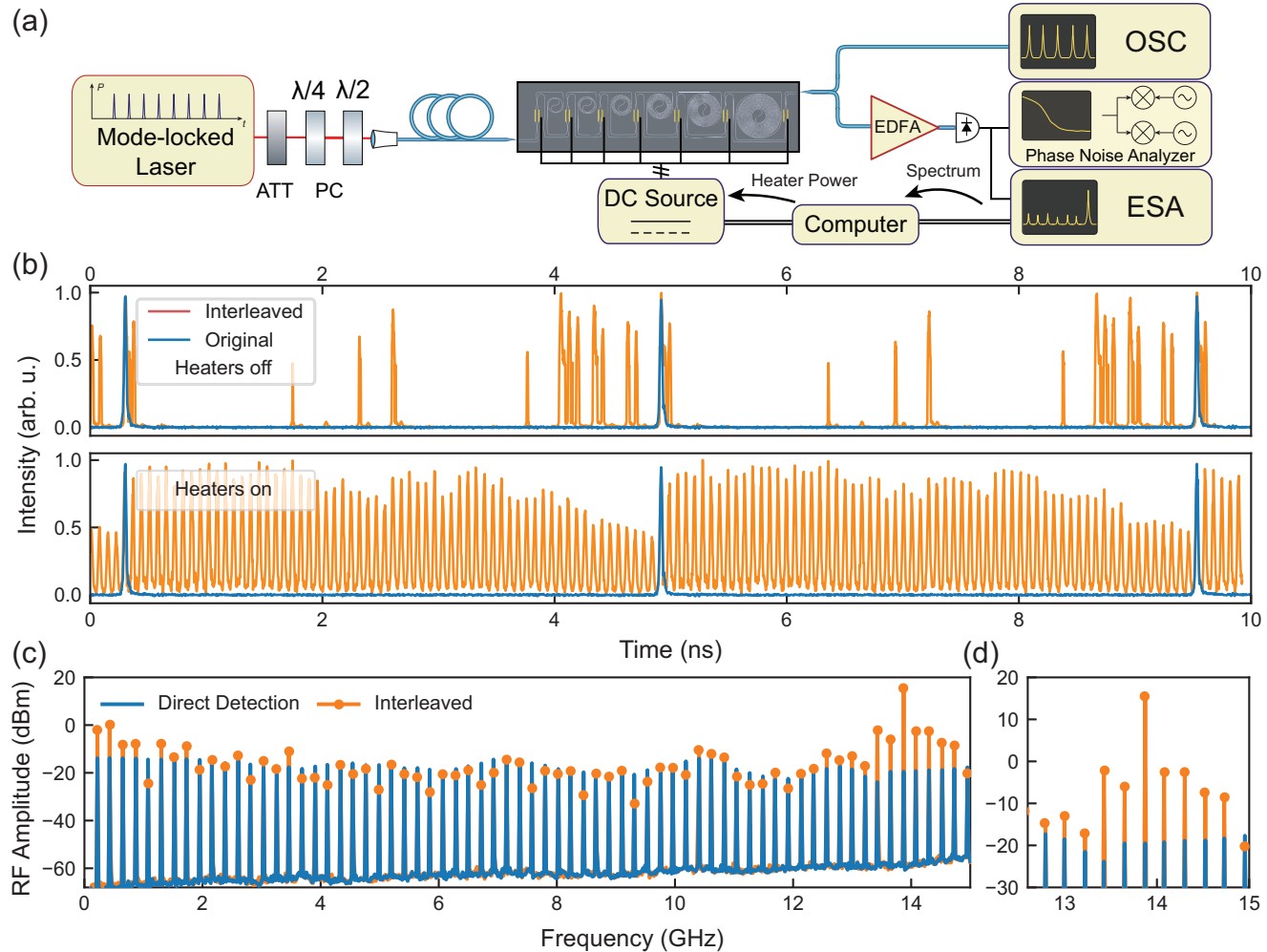

**Fig. 3 | Demonstration of 6-stage pulse interleaving using a femtosecond mode locked laser. a** The experiment setup for the pulse interleaving demonstration, where the 217 MHz pulse from the erbium-doped fiber-based mode-locked laser is interleaved. ATT attenuator, PC polarization controller, ESA radio frequency (RF) spectrum analyzer. **b** Time domain waveform of the pulses before and after interleaving, with the optimized heating power of the tunable couplers not applied (lower) and applied (upper). **c** Spectrum of the microwave signal generated by photodetection of the pulse train with and without interleaving, where the average optical power is set at 76 mW and 1.7 mW, respectively, for the maximum 13.9 GHz power in both cases. **d** Close-in view of the spectrum near the desired 13.9 GHz microwave, highlighting the 35 dB increase in the produced power.

ultra-low phase noise millimeter wave generation with a simple setup. The same design may also be implemented with 200 nm thick etched $Si_3N_4$ waveguides, which have a larger mode field diameter and can reduce nonlinear broadening. The subtractive fabrication process may also yield better control of the waveguide dimension, further improving the delay precision.

We demonstrate the pulse interleaving using 216.7 MHz repetition rate pulses from a free-running erbium-doped fiber-based MLL (OneFive Origami). After on-chip interleaving, the pulse train was directed to a MUTC PD (Freedom Photonics FP1015A, 5.3 V bias) for microwave generation. We optimized the heating power of the 7 tunable couplers to maximize the suppression of undesired harmonics on the RF spectrum (see Methods). Figure 3b illustrates the generation of 64 regular time-delayed copies of input pulses with optimal phase shift settings. The undesired harmonics on the RF spectrum could be suppressed by more than 14 dB (Fig. 3c) relative to the generated microwave level. The suppression ratio was limited by optical loss in long delay lines, which caused parasitic amplitude modulation of the pulse train (as seen in Fig. 3(b)). This effect results in sidebands at ± 433 MHz offset in the output microwave. We should note that this amplitude modulation doesn't degrade signal phase coherency and has minimal impact on applications.

Figure 4a shows the RF power as a function of the average optical power at the PD. The non-interleaved pulses saturated the PD at < 1mW average optical power and limited the generated 13.9 GHz harmonic power to < −19 dBm. The interleaved pulses could generate more than 16 dBm microwave power with 76 mW of average optical power when an erbium-doped fiber amplifier (EDFA) was used to amplify the signal before photodetection. This corresponds to >35 dB improvement in generated microwave power, only limited by the power handling of the photodetector. When a lower-cost PIN photodetector was used instead of the MUTC diode, >10 dB improvement in power was also achieved (Supplementary Note 5). The microwave phase noise was characterized by a phase noise analyzer (R&S FSWP) with a thermal noise-limited noise sensitivity at cross-correlation factor of 1000. As shown in Fig. 4(b), with the interleaver we reach the best phase noise performance of −160 dBc/Hz at 1 MHz offset for the generated −15 dBm 13.9 GHz microwave signal at -1.2 mA photocurrent. This noise floor is limited by the continuous wave shot noise and thermal noise floor[17] shown by the dashed line in Fig. 4b, and is -12 dB lower than the noise floor reached at the same photocurrent without using pulse interleaving. Here, the EDFA was not used to avoid extra noise introduced by pulse broadening induced by the Kerr nonlinearity in waveguides with tight optical mode confinement and spontaneous emission of the

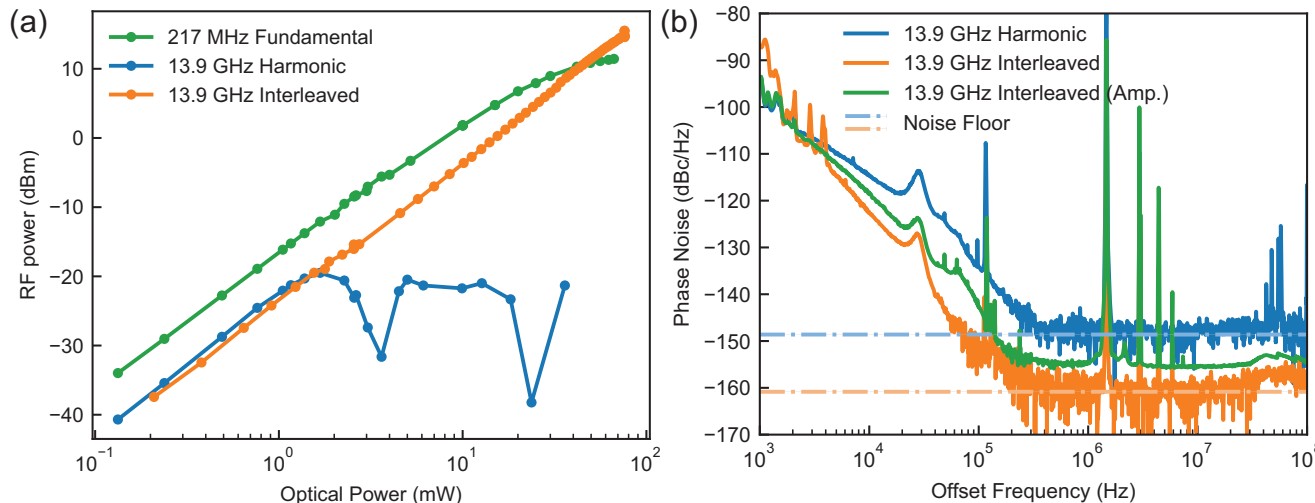

**Fig. 4 | Low-noise microwave generation using a large-scale photonic chip scale interleaver. a** The power of generated 13.9 GHz microwave as function of the average optical power after interleaving, which is shown together with the power of the 216 MHz fundamental radio frequency signal and the harmonic 13.9 GHz microwave generated by the pulse train without interleaving. This shows the saturation of the photodiode by the original pulse train and the higher power from the interleaved pulse train. **b** Phase noise of the generated 13.9 GHz microwave from the non-interleaved pulses, the interleaved pulses and the interleaved pulses amplified by an EDFA. The total shot and thermal noise floors are shown as dashed lines.

EDFA. When the output pulse train was amplified by the EDFA, the phase noise floor was slightly raised to −155 dBc/Hz at 16 dBm microwave power. At offset frequencies between 5 and 100 kHz, we also observed reduced phase noise in the interleaved pulse train. This phase noise advantage is likely due to the decreased phase noise converted from amplitude noise[13,30], as the photodiode was less saturated by the interleaved pulses with lower peak power. The elevated phase noise observed at low offset frequencies (approximately 1–5 kHz) occurs sporadically and does not reflect the intrinsic performance of the pulse interleaver. This intermittent noise likely originates from environmental perturbations affecting fiber-chip coupling stability and slight fluctuations in waveguide delays caused by external disturbances or noise from DC power sources. Such issues can be effectively mitigated by appropriately packaging the interleaver chip and employing stable, low-noise DC power sources, thus significantly enhancing overall system stability and performance.

In summary, we present a large-scale integrated pulse interleaver on a $Si_3N_4$ photonic integrated circuit platform. With low-loss waveguide delay lines, this 8.5 mm × 1.7 mm device can efficiently multiply the pulse repetition rate from 217 MHz to 14 GHz on a single chip. Applying this integrated photonic multiplier to a free-running femtosecond laser enabled an increase in the generated microwave power by 35 dB and a reduction of the phase noise floor by 12 dB. This integrated interleaver holds potential for miniaturized ultra-low-phase-noise photonic microwave generators with solid-state mode-locked lasers and can be co-integrated with future chip-based mode-locked lasers.

## Methods

### Details on sample fabrication

The photonic integrated circuits were fabricated with the photonic Damascene process with $SiO_2$ and $SiN_x$ hardmask for preform etching[26]. The waveguide width was reduced by 150 nm on the mask data to compensate for an offset introduced in the $SiN_x$ hardmask etching. For the microheaters, a 25 nm Ti adhesion layer and ~500 nm of Pt were sputtered on top of the photonic integrated circuit wafers with 3 μm of $SiO_2$ cladding. Direct-write ultraviolet lithography with 3 μm AZ ECI3027 photoresist was applied to define the 5 μm wide heater lines. We empirically find that the heater lines are more robust at high power when fabricated to be wide and thick. The heaters were etched with an Ar ion-beam etcher (Veeco Nexus IBE350) at −30° substrate

angle. The internal tracking number for the device used in the experiment is `D10701_F01C13` and the wafer characterized for uniformity in delay length is `D10602`.

### Details of the experimental setup for pulse interleaving

The interleaver can operate with the pulses entering either from the 6th stage or from the 1st stage. We sent the pulse to the 6th stage in the experiments to reduce the excessive nonlinear broadening of the pulses in the $Si_3N_4$ waveguides. In this way, the pulses were split before entering the longer delay lines, reducing the peak power and further mitigating nonlinear broadening (Supplementary Note 3). A 20 cm single-mode fiber delay line was used for pre-chirping the pulse (specified for 172 fs pulse duration) before coupling onto the interleaver chip. The time domain characterization of the resulting pulse was performed using a sampling oscilloscope with a 34 GHz optical input (Keysight 86105D), triggered by the synchronization output of the mode-locked laser.

### Automatic optimization of the splitting ratios

The power splitting ratios of the 7 MZI tunable couplers between the delay lines (Fig. 2a, b) were controlled by the heating power $|p_n|$ applied to one of the two arms, the sign of $p_n$ decides which arm is heated. The 14 heaters were driven by a multichannel DC source measure unit (nicslab). In order to find the optimal set points of all 14 heaters for microwave generation, where the power of the 64th harmonic is maximized while the other harmonics are suppressed, we monitored the RF spectrum of the photodetected pulses with an ESA (Keysight N9030A). We define a merit function $g = P_{64} - \sum_{n=1}^{63} P_n$, where $P_n$ is the power of the $n \times 217$ MHz harmonic in dBm unit, measured by the ESA. We employ the Nelder-Mead algorithm[31], as implemented in `scipy`, to optimize for the maximum $g$ as a function of $p_n$. The Nelder-Mead algorithm, being a derivative-free simplex optimization method, is well-suited for optimizing our experimentally measured, noisy target function. In our experiment setup with non-polarization-maintaining fiber, the polarization of the beam coupled on chip may not be perfectly aligned with the chip plane (for TE mode coupling). The misalignment also affects the extinction ratio and the merit function. For this, the initial automatic optimization was performed iteratively with manual fine tuning of the fiber polarization controller. We characterized the on-chip optical loss after the optimization of polarization and the

application of the optimal heater setting. The 4.7 dB fiber-to-chip coupling loss and the 3 dB loss caused by the equal splitting of the output power from the two ports are subtracted.

## Data availability

The data generated in this study as well as related processing and simulation code have been deposited in the `Zenodo` repository under accession code https://doi.org/10.5281/zenodo.15162735.

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

## Acknowledgements

The samples were fabricated in the EPFL center of Micro-NanoTechnology (CMi). This work was supported by the EU H2020 research and innovation program under grant No. 965124 (FEMTOCHIP).

## Author contributions

Z.Q. and N.S. performed the experiments. Z.Q., N.S., and Y.L. carried out data analysis and simulations. Y.L. and X.J. provided experimental supports. Z.Q., Y.L and N.S. designed $Si_3N_4$ photonic integrated circuits. R.N.W., Z.Q. and X.J. fabricated the $Si_3N_4$ samples. Z.Q. wrote the manuscript with the assistance from N.S. and Y.L., with the input from all co-authors. F.X.K. and T.J.K supervised the project.

## Competing interests

T.J.K. is a cofounder and shareholder of LiGenTec SA, a start-up company offering $Si_3N_4$ photonic integrated circuits. The remaining authors declare no competing interests.
