## [Transparent Peer Review file · Nature Communications]

Large-scale photonic chip based pulse interleaver for low-noise microwave generation

Corresponding Author: Professor Tobias Kippenberg

Version 0:

Reviewer comments:

Reviewer #2

(Remarks to the Author)

In this paper entitled "Large-scale photonic chip based pulse interleaver for low-noise microwave generation," the authors successfully fabricated a pulse interleaver that can be implemented on a Si_3N_4 photonic integrated circuit platform, and they generated low-noise microwaves at 14 GHz from a mode-locked laser with a repetition rate of 217 MHz. In the past, such experiments required pulse interleavers built using free-space optics or optical fibers, but in this study, the authors integrated a six-stage pulse interleaver on a very compact chip, thereby realizing an ultra-compact low-noise microwave generator. The paper covers fundamental information ranging from silicon photonics fabrication to the evaluation of the generated microwave signal, and overall, it is very well written. This work represents a significant step toward applying laboratory-level OFD technology to real-world applications. I would generally recommend publication in Nature Communications; however, since the paper contains highly specialized content, it might be beneficial to refine the introduction so that the impact of this research is more accessible to researchers outside the field of OFD. Additionally, I have summarized some other points of concern below.

- The introduction presents several state-of-the-art microwave generation studies and provides specific phase noise values. However, for a general audience, it might be somewhat difficult to understand the significance of these values and how superior the authors' results are. It would be helpful, if possible, to include more specific application examples (e.g., mentioning that certain applications require phase noise below a specific threshold).
- The main text states that the waveguide length can be controlled to within less than 2 ps. With this level of manufacturing error, up to what repetition frequency can be tolerated (for instance, can one estimate the frequency range over which an improvement of more than 30 dB can be achieved)?
- In relation to the above, what is the degree of error among different chips fabricated on the same wafer?
- A comment on the long-term stability of the interleaver is desired. For example, how frequently is the optimization (as described in the Methods) updated? Is continuous operation possible, and is temperature control implemented for the Si_3N_4 substrate?
- In Figure 1(b), the orientation of the four PBSs (if that is indeed what they are) appears to be reversed.
- In Figure 4, parts (a) and (b) are missing.
- In Figure 4(b), in the plot comparing the cases with and without an EDFA, why is the phase noise at low frequencies (1 kHz ~ 5 kHz) higher in the absence of the EDFA? Although there is a comment on the noise in this region at the end of Section 3, it does not seem to adequately explain why using the EDFA results in reduced noise.
- Regarding Figure 4(b), what is the cause of the large peak observed at 1.5 MHz or its harmonics?

Version 1:

Reviewer comments:

Reviewer #2

(Remarks to the Author)

After reviewing the authors' detailed rebuttal letter, I am pleased to confirm that all the concerns raised in my initial review have been satisfactorily addressed.

In the introduction section, although the phase noise requirements vary depending on the application and cannot be generalized, the authors have added specific examples to make the content engaging for a broader audience.

The authors have also analyzed the delay-length error and demonstrated that, even with the current error level, no performance degradation is observed in frequency generation up to 55 GHz using an 8-stage pulse interleaver. Additionally, further evaluations were conducted to determine specific standard deviation values, indicating that a high yield can be expected.

Finally, regarding the increased noise observed at the low-frequency side of the phase noise spectrum, analysis of multiple measurement results revealed that this phenomenon occurs only sporadically. The manuscript has been updated to include a discussion on this point.

We sincerely thank the reviewer for their thorough and thoughtful evaluation of our manuscript. We greatly appreciate the constructive feedback, which has helped us improve the clarity, rigor, and overall quality of the work. In response to the reviewer's comments, we have carefully revised the manuscript.

In the following, we provide a detailed, point-by-point response to the reviewer's comments. For clarity and ease of reading, the reviewer's original comments are shown in black, our responses are provided in blue, and the corresponding actions taken for the manuscript are highlighted in red.

Reply to comments from Reviewer #2

In this paper entitled "Large-scale photonic chip based pulse interleaver for low-noise microwave generation," the authors successfully fabricated a pulse interleaver that can be implemented on a Si_3N_4 photonic integrated circuit platform, and they generated low-noise microwaves at 14 GHz from a mode-locked laser with a repetition rate of 217 MHz. In the past, such experiments required pulse interleavers built using free-space optics or optical fibers, but in this study, the authors integrated a six-stage pulse interleaver on a very compact chip, thereby realizing an ultra-compact low-noise microwave generator. The paper covers fundamental information ranging from silicon photonics fabrication to the evaluation of the generated microwave signal, and overall, it is very well written. This work represents a significant step toward applying laboratory-level OFD technology to real-world applications. I would generally recommend publication in Nature Communications; however, since the paper contains highly specialized content, it might be beneficial to refine the introduction so that the impact of this research is more accessible to researchers outside the field of OFD. Additionally, I have summarized some other points of concern below.

- The introduction presents several state-of-the-art microwave generation studies and provides specific phase noise values. However, for a general audience, it might be somewhat difficult to understand the significance of these values and how superior the authors' results are. It would be helpful, if possible, to include more specific application examples (e.g., mentioning that certain applications require phase noise below a specific threshold).

We sincerely appreciate the reviewer's insightful comments regarding the accessibility of our manuscript for a broader audience. Following the reviewer's suggestion, we have enhanced the introduction by explicitly highlighting two relevant application examples—wireless telecommunications and radar systems—supported by appropriate references, to clearly demonstrate the practical significance of achieving low-phase-noise microwaves. We believe these additions will help readers from diverse backgrounds better appreciate the impact of our research.

We would also like to respectfully clarify the challenge of specifying a universally applicable threshold for phase noise. Determining such thresholds involves numerous variables, including specific system design parameters and operating environments. For instance, as detailed in reference [15] of our revised manuscript, a very-low phase noise local oscillator in the transmitter is shown to significantly reduce interference to close by communications, thus enhancing spectrum utilization efficiency. However, exact phase noise requirements depend strongly on factors such as the chosen spectrum allocation strategy, channel spacing, and transmission power, all of which vary considerably across different scenarios.

Similarly, reference [13] discusses radar applications and emphasizes that while low-phase-noise oscillators undoubtedly provide advantages, precisely quantifying these benefits typically necessitates extensive Monte-Carlo simulations tailored explicitly to the given FMCW radar configurations. Additionally, for analog-to-digital conversion in instrumentation systems, there is a clear and ongoing trend toward higher sampling rates and improved resolution (effective number of bits), which underscores the critical need for reduced timing jitter. Yet again, the specific jitter requirements vary significantly depending on the implementation context.

Given these inherent complexities and variations across different applications, specifying universally applicable numerical phase noise thresholds is beyond the scope of the current study. Nevertheless, we have carefully refined the manuscript's introduction to clearly articulate the detrimental impact of phase noise on system performance and to direct interested readers toward relevant literature, allowing them to determine appropriate phase noise thresholds for their specific applications.

Action taken

We have revised and improved the manuscript's introduction to enhance accessibility for readers outside our immediate research field by incorporating illustrative application examples and clear explanations regarding the impact of phase noise on system performance.

- The main text states that the waveguide length can be controlled to within less than 2 ps. With this level of manufacturing error, up to what repetition frequency can be tolerated (for instance, can one estimate the frequency range over which an improvement of more than 30 dB can be achieved)?

We sincerely thank the reviewer for raising this important point regarding the maximum repetition frequency given our fabrication precision. As indicated in our manuscript, the waveguide delay lengths are controlled within an accuracy better than 2 ps, and we agree that quantifying the impact of this fabrication precision on achievable microwave frequencies and associated noise reduction is important.

Two primary factors limit phase-noise enhancement performance when there is a finite delay-length error: firstly, a reduction in the generated microwave signal power, which negatively affects the phase noise floor due to lower signal-to-noise ratio (SNR); secondly, degradation of the beneficial shot-noise correlation reduction achievable with short optical pulses. Both of these effects have been comprehensively analyzed in the reference [29] "Optical amplification and pulse interleaving for low-noise photonic microwave generation," *Optics Letters*, 39(6), 2014. Their study demonstrated that while the microwave signal magnitude is relatively insensitive to delay-length errors (even up to 5 ps at 10 GHz), significant deterioration of phase noise occurs with increasing delay-length errors.

To quantitatively assess the impact of our fabrication accuracy (within 2 ps) on performance, which will become more significant at higher harmonic frequencies, we have performed numerical simulations modeling realistic conditions. In these simulations, we considered practical waveguide losses (7 dB/m) and a first-stage delay length of 32.9769 cm. The simulations evaluated the phase-noise improvement achievable via shot-noise correlation and the power of generated microwave, following the approach in "Analysis of shot noise in the detection of ultrashort optical pulse trains," *JOSA B*, 30(6), 2013. The fabrication error is modeled to be proportional to the delay length and the error in the longest stage is used in the representation.

In Figure R1(a), we illustrate the simulated degradation of harmonic microwave power as a function of delay-length error for pulse interleavers with 6, 7, and 8 stages, corresponding to microwave frequencies of 13.9 GHz, 27.8 GHz, and 55.7 GHz, respectively. The results clearly show that, even for the highest frequency considered (55.7 GHz), the intensity of the generated microwave signal remains largely unaffected by delay-length errors up to our fabrication tolerance of 2 ps. Thus, the full SNR-driven phase noise improvement is expected to be maintained at this precision level.

Additionally, Figure R1(b) demonstrates the calculated phase-noise reduction achievable through shot-noise correlation for the same frequency range. Our simulation indicates that, even at the highest frequency of 55.7 GHz, a 2 ps delay-length error still allows for more than 5 dB of shot-noise correlation improvement.

We would like to note there are other reductions by AM-to-PM noise reduction will be provided by interleaving and our simulations assume ideal photodiode performance. Precisely predicting the amount of SNR improvement is much more difficult as this requires detailed modeling of the saturation in the photodiode. In practice, additional complexities such as carrier scattering dynamics within modified uni-traveling carrier (MUTC) photodiodes (as detailed by Sun et al., "Broadband noise limit in the photodetection of ultralow jitter optical pulses," *Physical Review Letters*, 113(20), 2014) would reduce the achievable shot-noise correlation improvement with short pulses. Consequently, the actual performance impact of delay-length errors could be smaller than our idealized simulations suggest. Temporal broadening by the dispersion in the delay lines may also degrade the shot noise correlation for a short <300 fs input pulse, but for the longer pulses that are still benefiting from the correlation despite carrier scattering, the impact is much smaller.

In summary, given our demonstrated fabrication accuracy of better than 2 ps, our simulations confidently support achieving a substantial phase-noise improvement at microwave frequencies up to 55.7 GHz, corresponding to an 8-stage pulse interleaver.

Figure R1 (a): Simulated power of generated microwave power as a function of the delay error in the longest delay coil, shown for cases with 6, 7 and 8 stages of interleaving. (b) Simulated phase noise reduction by short-noise correlation as a function of the delay error.

- In relation to the above, what is the degree of error among different chips fabricated on the same wafer?

We thank the reviewer for raising this important question. In response, we have performed additional characterizations to quantify the variation of delay lengths across multiple devices fabricated on the same 4-inch wafer. Utilizing the measurement and analysis method

described in our supplementary information, we conducted Fourier transform-based delay characterization for the 9 identical devices produced in the 9 stepping fields on the wafer.

The except of the normalized Fourier transformed spectrum concerning the longest stage is shown below as Figure R1(a). By extracting the peak position, we found that the peak-to-peak variation is 2 ps and the standard deviation is 0.57 ps, which corresponds to 0.025% of the delay. This measured variability is comparable to the 2-ps precision previously demonstrated in the device discussed in the main manuscript, confirming that our fabrication process achieves consistent and reproducible results suitable for volume manufacturing.

The geometrical distribution of error across the wafer is shown as Figure R1(b). We believe the majority of the variation is caused by the waveguide height difference induced in the chemical mechanical polishing process in the current photonic Damascene process and can be further reduced by changing to a “substrative” fabrication process.

Figure R2: (a) Fourier transform of the 9 measured transmission spectrum showing the delay of the longest stage. (b) Wafer map of the extracted delay error.

Action taken

We have explicitly included the standard deviation of the delay length measured across different stepping fields on the same wafer (approximately 0.57 ps), demonstrating that the fabrication process maintains high precision across the wafer. This highlights that our fabrication method is scalable and capable of achieving high yields in volume manufacturing.

- A comment on the long-term stability of the interleaver is desired. For example, how frequently is the optimization (as described in the Methods) updated? Is continuous operation possible, and is temperature control implemented for the Si_3N_4 substrate?

We thank the reviewer for raising this important question regarding the long-term operational stability of our integrated pulse interleaver. In our experimental demonstration, which was conducted in an environmentally controlled laboratory setting, we optimized the device only once at the initial setup, as described in the Methods section. Subsequently, no further optimization was required throughout the duration of our measurements (around 20 minutes).

Currently, no active temperature control was implemented due to mechanical constraints associated with the chip mount used in our experiment. However, we anticipate that implementing temperature stabilization and further improving the system design will significantly enhance the long-term operational stability and robustness of the interleaver.

- In Figure 1(b), the orientation of the four PBSs (if that is indeed what they are) appears to be reversed.

We thank the reviewer for pointing out this mistake in the illustration. We have corrected it in the revised version of the manuscript.

Action taken

Corrected the orientation of the four PBSs in the Figure 1(b).

- In Figure 4, parts (a) and (b) are missing.

We thank the reviewer for pointing out this mistake in the manuscript preparation. We added the panel labels in the revised version of the manuscript.

Action taken

Edited the figure to add labels for panel (a) and (b).

- In Figure 4(b), in the plot comparing the cases with and without an EDFA, why is the phase noise at low frequencies (1 kHz ~ 5 kHz) higher in the absence of the EDFA? Although there is a comment on the noise in this region at the end of Section 3, it does not seem to adequately explain why using the EDFA results in reduced noise.

We appreciate the reviewer raising this insightful point regarding the increased phase noise at low offset frequencies (1 kHz to 5 kHz) in Figure 4(b) when the EDFA is not used. Upon careful re-examination of our data, we attribute this elevated low-frequency noise primarily to sporadic technical perturbations rather than fundamental device characteristics. These perturbations likely include environmental influences such as acoustic vibrations affecting fiber-to-chip coupling efficiency, RF interference within the shared laboratory environment, and fluctuations originating from the mode-locked laser or the DC power sources used for the microheaters.

To further investigate this issue, we revisited multiple measurements conducted under identical conditions without the EDFA. The results (presented in Figure R3) clearly illustrate that the elevated low-frequency noise appears only sporadically in certain datasets, confirming its non-intrinsic and intermittent nature. We observed no systematic correlation between EDFA usage and low-frequency noise levels, suggesting the difference observed in Figure 4(b) is coincidental rather than fundamental to the interleaver's operation.

In summary, the observed low-frequency noise difference is not fundamentally linked to EDFA usage but results from intermittent external factors during measurements.

Action taken:

We have updated the manuscript's discussion regarding the low-frequency phase noise observed in Figure 4(b), clarifying its sporadic nature and emphasizing practical strategies (such as device packaging and improved electrical supply stability) to minimize these environmental effects.

Figure R3: Three phase noise datasets taken consecutively when the EDFA is not in use.

- Regarding Figure 4(b), what is the cause of the large peak observed at 1.5 MHz or its harmonics?

This peak likely originates from the mode-locked laser itself, from the characterization system or from picked-up RF interference in the shared laboratory. We have observed the same peak appearing on the characterized phase noise without interleaving (as shown as the blue trace in Figure 4(b) in the manuscript) as well as the phase noise of the fundamental tone without interleaving.

Reply to comments from Reviewer #2

After reviewing the authors' detailed rebuttal letter, I am pleased to confirm that all the concerns raised in my initial review have been satisfactorily addressed.

In the introduction section, although the phase noise requirements vary depending on the application and cannot be generalized, the authors have added specific examples to make the content engaging for a broader audience.

The authors have also analyzed the delay-length error and demonstrated that, even with the current error level, no performance degradation is observed in frequency generation up to 55 GHz using an 8-stage pulse interleaver. Additionally, further evaluations were conducted to determine specific standard deviation values, indicating that a high yield can be expected.

Finally, regarding the increased noise observed at the low-frequency side of the phase noise spectrum, analysis of multiple measurement results revealed that this phenomenon occurs only sporadically. The manuscript has been updated to include a discussion on this point.

We sincerely thank the reviewer for carefully evaluating our detailed response and revised manuscript. We greatly appreciate the reviewer's thorough and constructive feedback, which significantly improved the clarity and accessibility of our paper. We are pleased to hear that our revisions have adequately addressed all concerns, including refining the introduction with specific application examples, providing additional analysis on delay-length errors and their implications for high-frequency operation, and clarifying the origin and nature of the excessive low-frequency noise.